# Integrating human activity into food environments can better predict cardiometabolic diseases in the United States

Ran Xu[1,2], Xiao Huang [3], Kai Zhang[4], Weixuan Lyu[5], Debarchana Ghosh[2,5], Zhenlong Li [6] & Xiang Chen [2,5] ✉

The prevalence of cardiometabolic diseases in the United States is presumably linked to an obesogenic retail food environment that promotes unhealthy dietary habits. Past studies, however, have reported inconsistent findings about the relationship between the two. One underexplored area is how humans interact with food environments and how to integrate human activity into scalable measures. In this paper, we develop the retail food activity index (RFAI) at the census tract level by utilizing Global Positioning System tracking data covering over 94 million aggregated visit records to approximately 359,000 food retailers across the United States over two years. Here we show that the RFAI has significant associations with the prevalence of multiple cardiometabolic diseases. Our study indicates that the RFAI is a promising index with the potential for guiding the development of policies and health interventions aimed at curtailing the burden of cardiometabolic diseases, especially in communities characterized by obesogenic dietary behaviors.

Cardiometabolic diseases (CMDs) are prevalent in the United States (US). The most recent data shows that 41.9%[1], 47%[2], and 38.1%[3] of the adults in the US are obese, have hypertension, and have high cholesterol, respectively. Rising levels of CMDs in the country are inseparable from structural changes in the food systems, including economic policies which drove the rapid expansion of fast food chains, technical advances that popularized ultra-processed foods, and the exurbanization process that decentralized residential communities[4]. These policy and urban changes ultimately induced an uneven foodscape with detrimental health consequences, primarily the development of CMDs[5–12]. As such, national food and health initiatives have taken strides to evaluate food access disparities using food environment measures, including the Food Access Research Atlas and the modified retail food environment index (mRFEI)[13], as evidence when designing policy interventions to improve diet-related health outcomes[14].

However, the connection between such food environment measures and CMDs remains far from conclusive[5,15]. For example, while some studies identified a positive relationship between fast food restaurant accessibility and obesity[6–9], others found negative[10], null[11], and mixed relationships[12]. Numerous confounding factors may account for such inconsistencies, such as measures of environmental exposures, obesity indicators, and units of analysis[2]. One often overlooked aspect is human mobility[16]. Specifically, food environment measures usually rely on a fixed number of retailers within predefined administrative units (e.g., counties, census tracts), assuming that people are exclusively exposed to food retailers within a unit[5]. In reality, however, consumers' food procurement activities often extend beyond a single location or a unit[17]. Further, because of tiers of behavioral uncertainties (e.g., food culture, health education, food security status), their food exposure and activities may not mirror the foodscape near their place of residence. These behavioral uncertainties play a more direct role

[1]Department of Allied Health Sciences, University of Connecticut, Storrs, CT 06269, USA. [2]Institute for Collaboration on Health, Intervention, and Policy (InCHIP), University of Connecticut, Storrs, CT 06269, USA. [3]Department of Environmental Sciences, Emory University, Atlanta, GA 30322, USA. [4]Department of Environmental Health Sciences, School of Public Health, University at Albany, State University of New York, Rensselaer, NY 12144, USA. [5]Department of Geography, University of Connecticut, Storrs, CT 06269, USA. [6]Geoinformation and Big Data Research Lab, Department of Geography, University of South Carolina, Columbia, SC 29208, USA. ✉e-mail: xiang.chen@uconn.edu

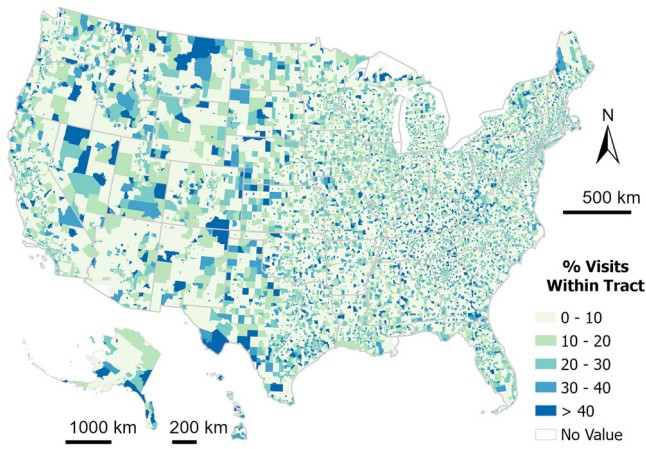

**Fig. 1 | GPS-based food retailer visits in the US.** The map shows the percentage of visits to food retailers in 2018–2019 within a ½ mile boundary of the residents' home census tract. The map was made in ESRI ArcGIS Pro 3.0.2.

than the proximity to retailer locations in shaping store patronization, food choices, and eventually, diet-related health outcomes[18–20].

Integrating human activity into the study of food environments paves the way towards better understanding of the widening health disparities in the US. It has been found that racial and ethnic minorities, particularly non-Hispanic Blacks and Hispanics, face the highest risks of CMDs[21]. The adverse diet-related health outcomes faced by minorities extend beyond an unequal foodscape and are fundamentally induced by structural barriers, such as financial resources and personal transportation, that restrict marginalized populations from accessing nutritious food and health resources[22]. While it is impossible to enumerate all structural barriers that shape food access, a better understanding of human mobility as a crucial manifestation of the resources attainable to individuals will help further demystify diet-related health disparities[23]. This causal pathway to health disparities has been acknowledged, but it has yet to be substantiated with broad-scale human mobility data and activity-integrated models.

The last decade has witnessed a growing research focus on human mobility by exploring how individuals' food procurement trips give rise to different food consumption patterns and risks of CMDs[18,24–28]. However, most of the existing studies, especially those utilizing Global Positioning System (GPS) devices, have a limited spatiotemporal scale due to the nature of active data collection. Besides, conclusions derived from regional case studies usually cannot be generalized to inform policymaking on a broader scale. Expanding the observation scale of human mobility, developing activity-based measures, and justifying their links to risks of CMDs has yet to be fully explored.

In this study, we leverage a large-scale human mobility dataset garnered from GPS-enabled mobile devices, covering over 94 million aggregated visit records to roughly 359,000 food retailers across the US for two years. Fig. 1 presents the preliminary result of the data focusing on food retailer visits, showing that the majority of the food retailer visits were beyond residents' immediate neighborhoods. This mobility pattern signifies the importance of integrating human activity into the study of food environments, as analyzing household or retailer locations by census tract fall short in capturing the majority of food activities. Based on this dataset, we construct a retail food activity index (RFAI) on a granular spatial scale (i.e., census tract), and then justify its validity by comparing it with a representative location-based food environment index in terms of the strength of associations with various cardiometabolic health outcomes. This comparison demonstrates that the RFAI can better predict the prevalence of multiple CMDs than the location-based food environment index. By performing a nationwide food retailer visit assessment, this study can offer valuable insights for policymakers seeking to devise contextualized food policy initiatives and health intervention strategies for communities in need of behavioral changes.

## Results

### Food retailer visit patterns

We constructed food retailer visit patterns based on a national human mobility dataset over a two-year period (2018–2019), which was aggregated from a large sample of GPS-enabled mobile devices in the US and contained information on the number of visits to each point of interest (POI) and visitors' home census tract (see more details in methods). We included visit data to five categories of food retailers, including supermarket and grocery stores, warehouse clubs, fruit and vegetable markets, limited-service restaurants, and convenience stores. Fig. 1 depicts the percentage of visits to food retailers that were within a ½ mile boundary of each resident's home census tract. The selection of the food retailer categories and the buffer distance was consistent with the modified retail food environment index (mRFEI), which is a widely used location-based food environment index[13]. Fig. 1 shows that the majority of the food retailer visits were beyond residents' immediate neighborhoods. On average, only 20.8% (standard deviation [SD] = 13.9%) of food retailer visits were within a half-mile boundary of the residents' home census tracts. Residents traveled a median distance (calculated from centroid of the census tract to each food retailer) of 3.70 miles to food retailers (interquartile range [IQR] = 2.61–7.02), and among all visits, 12.8% (SD = 13.9%) were within one mile, 37.1% (SD = 19.4%) were within one to five miles, 16.8% (SD = 12.8%) were within five to ten miles, 12.8% (SD = 12.5%) within ten to twenty miles, and 20.5% (SD = 14.8%) were beyond twenty miles.

Food retailer visit patterns can exhibit variations by urban status and across different sociodemographic groups. Results from regression analyses showed that multiple sociodemographic characteristics of the census tract were associated with the median distance residents traveled (Supplementary Fig. 1 and Table 3). The strongest predictor was population density, such that a 100% increase in the population density of residents' home census tracts was associated with a 27.6% (95% confidence interval [CI] 27.4–28.0) decrease in the median distance traveled. In addition, urban residents, on average, traveled 25.5% (95% CI 24.2–26.9) shorter distances than residents in non-urban areas. A one percentile increase in the social vulnerability of household composition and disability, minority status and language, and housing type and transportation, was associated with a 0.055% (95% CI 0.041–0.070) decrease, 0.249% (95% CI 0.234–0.264) increase, and 0.164% (95% CI 0.151–0.177) decrease in distance traveled, respectively. Further, an investigation by racial and ethnic groups showed that, conditional on other variables, residents in predominantly non-Hispanic White census tracts (defined as the respective population ≥ 50%) traveled 15% (95% CI 14–16.1) shorter distances than others, while residents in predominantly non-Hispanic Black census tracts traveled 3.5% (95% CI 2.1–4.9) further than others.

### Retail food activity index

Given that most of the visits were beyond residents' home tracts, we created the retail food activity index (RFAI). The index, representing the percentage of visits to healthy food retailers for residents living in a given census tract, ranges from 0 to 100 (where 0 means the lowest level of healthy food retailer visits). We compared it with the mRFEI, which is defined as the percentage of healthy food retailers located in a census tract[13]. Fig. 2 depicts the spatial distributions of the RFAI (activity-based) and the mRFEI (location-based)[13]. There was little consistency between the two indices, as evidenced by a low correlation level (correlation coefficient = 0.069, 95% CI 0.061–0.076). The Supplementary Figs. 2 and 3 reveal further statistical insights into their distributions.

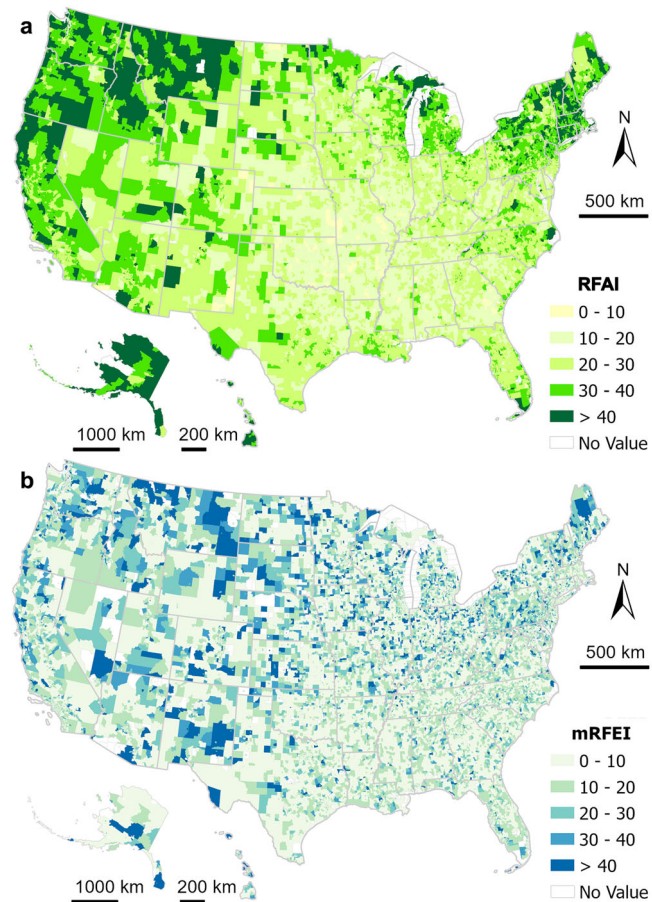

**Fig. 2 | US census tract level maps for two food environment indices.** The maps show the spatial distributions of **a** RFAI as the activity-based food environment index and **b** mRFEI as the location-based food environment index. The maps were made in ESRI ArcGIS Pro 3.0.2.

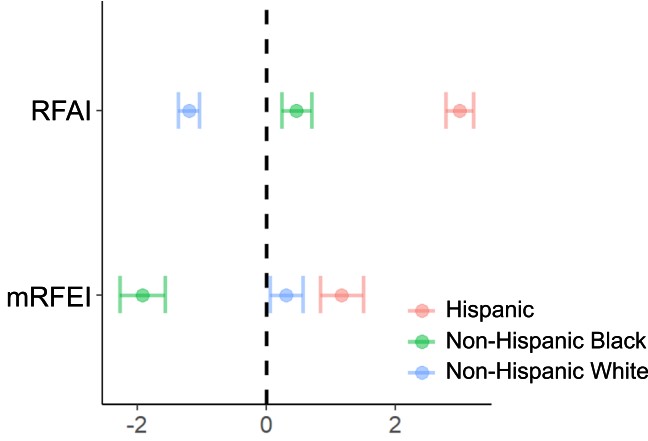

**Fig. 3 | Estimated differences in RFAI and mRFEI between census tracts with a predominantly racial/ethnic group and other census tracts (adjusted for covariates).** Center of the error bar represents the estimated mean difference and the error bar represents the 95% CI. For mRFEI, $n$ = 6,679 census tracts for predominantly Hispanic census tracts, $n$ = 5727 census tracts for predominantly Non-Hispanic Black census tracts, and $n$ = 45,876 census tracts for predominantly Non-Hispanic White census tracts. For RFAI, $n$ = 6,688 census tracts for predominantly Hispanic census tracts, $n$ = 5,732 census tracts for predominantly Non-Hispanic Black census tracts, and $n$ = 46,238 census tracts for predominantly Non-Hispanic White census tracts.

We performed regression analyses to examine the associations between sociodemographic characteristics and the RFAI at the census tract level. Results showed that census tracts with a lower RFAI were more socioeconomically deprived. Specifically, a one percentile increase in the social vulnerability of socioeconomic status as well as housing type and transportation was associated with 0.0581 (95% CI 0.055–0.061) and 0.023 (95% CI 0.021–0.025) units lower RFAI, respectively. Census tracts with a food desert status (defined as "low-income, low-access" census tracts by United States Department of Agriculture [USDA][29]) had 1.499 (95% CI 1.355–1.644) units lower RFAI. Demographically, a 100% increase in the population density of the residents' home census tracts was associated with 0.569 (95% CI 0.526–0.611) units higher RFAI. We also examined these relationships for the mRFEI. The results showed that while the directions of the relationships were consistent between the mRFEI and the RFAI for socioeconomic status and food desert status, the mRFEI exhibited weaker or opposite relationships with many other sociodemographic factors (see details in Supplementary Table 4).

Investigation of the racial and ethnic groups revealed that a one percentile increase in social vulnerability of minority status was associated with 0.056 (95% CI 0.053–0.058) units higher RFAI. Fig. 3 presents the results for census tracts predominantly inhabited by a specific racial and ethnic subgroup. It shows that predominantly Hispanic census tracts had significantly higher RFAI (2.992 units higher, 95% CI 2.773–3.209) than other census tracts, followed by non-Hispanic Black (0.463 units higher, 95% CI 0.231–0.694), while predominantly non-Hispanic White census tracts had significantly lower

RFAI (1.199 units lower, 95% CI 1.035–1.363). This pattern was different for the mRFEI. While predominantly Hispanic census tracts still had higher mRFEI (1.166 units, 95% CI 0.834–1.497) than other census tracts, predominantly non-Hispanic Black census tracts were 1.919 (95% CI 1.567–2.271) units lower and predominantly non-Hispanic White census tracts were 0.309 (95% CI 0.060–0.559) units higher in the mRFEI than others.

### Associations with the prevalence of CMDs

To explore the health implications of the study, we investigated the associations between the RFAI and the prevalence of five CMDs at the census tract level, as shown in Fig. 4 (see Supplementary Fig. 4 for spatial distributions of individual CMD variables). We also compared these associations with that of the mRFEI, which was commonly used to study correlations between food environments and diet-related health outcomes[30,31]. The results showed that the RFAI had stronger associations with the prevalence of obesity, high cholesterol, and high blood pressure than the mRFEI. Specifically, after adjusting for covariates, one interquartile increase (i.e., 25th percentile to 75th percentile) in the RFAI was associated with 0.629% (95% CI 0.585–0.674) lower prevalence of obesity, 0.171% (95% CI 0.135–0.206) lower prevalence of high cholesterol, and 0.521% (95% CI 0.471–0.571) lower prevalence of high blood pressure, respectively. On the contrary, the association between the mRFEI and the prevalence of CMDs were much weaker, such that one interquartile increase in the mRFEI was only associated with 0.174% (95% CI 0.143–0.204) lower obesity, 0.064% (95% CI 0.039–0.089) lower high cholesterol, and 0.261% (95% CI 0.226–0.295) lower high blood pressure, respectively. Further sensitivity analyses showed that the effects of the RFAI on predicting the prevalence of obesity, high cholesterol, and high blood pressure were largely linear and robust to various alternative specifications (especially for obesity and high blood pressure), including comparisons against a more updated location-based food environment index (Supplementary Fig. 5), performing county-level analysis (Supplementary Fig. 6), accounting for spatial autocorrelations (Supplementary Fig. 7), examining non-linear effects (Supplementary Fig. 8), and examining the relationships by racial/ethnic groups (Supplementary Fig. 9).

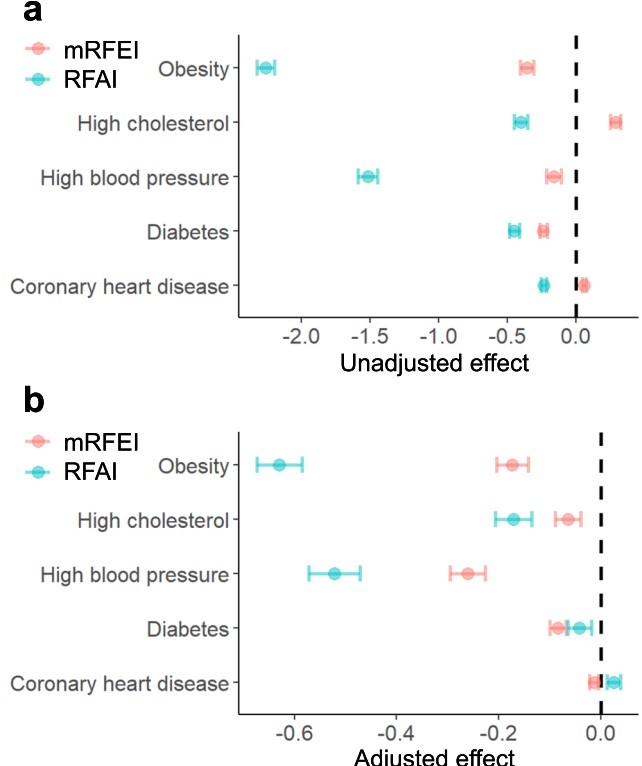

**Fig. 4 | Associations between two food environment indices and CMDs.** Estimated change (%) and 95% CI in the prevalence of five CMDs associated with one interquartile increase in the RFAI or mRFEI (i.e., 25th percentile to 75th percentile) based on **a** unadjusted census tract level model ($n$ = 69,695 census tracts for mRFEI and $n$ = 70,270 for RFAI) and **b** models adjusted for covariates ($n$ = 67,856 for mRFEI and $n$ = 68,260 for RFAI). Center of the error bar represents the estimated change (%) and the error bar represents the 95% CI.

## Discussion

Location-based food environment measures (e.g., mRFEI[13]) and related tools (e.g., Food Access Research Atlas[29]) have been long employed for evidencing the inequality of foodscapes and understanding their effects on diet-related health outcomes. However, it remains unclear that location-based food accessibility is a significant predictor of CMDs in many cases[6–9], and this ambiguity leads to questions about its reliability to inform policymaking. We argue that location-based food environmental studies are missing an important behavioral component, food procurement activity, which is a precedent to diets and eventually diet-related health outcomes. Using a large-scale GPS-based human mobility dataset in the US, we showed that the majority of food retailer visits extended beyond residents' immediate neighborhoods. We then proposed an aggregate measure that integrates food retailer visits, the RFAI, and demonstrated that it had a weak correlation with the widely used location-based food environment index (mRFEI) and exhibited significant differences in terms of their associations with many other important sociodemographic factors. More importantly, we showed that the RFAI can better predict the prevalence of multiple CMDs, including obesity, high blood pressure, and high cholesterol, and this conclusion is robust to adjusting for various socio-demographic factors and multiple alternative model specifications. For example, across models, we found that the association between the RFAI and obesity prevalence was about three times stronger than that between the mRFEI and obesity. To our best knowledge, this is the first nationwide study to demonstrate the value of integrating human mobility into the study of food environment on a granular scale in the US.

Many of our results are consistent with previous studies. The finding that residents traveled longer distances for food in low-population density areas and non-urban areas was consistent with the previous studies[20,32,33]. The finding can be explained by the consolidation of large food retailers in urban areas or regional hub towns as a result of urban sprawl, which has left rural areas with barely any quality food access[34]. Also, previous literature suggests that areas with a low socioeconomic status or low food access are also more likely to have less healthy food environments or be associated with less healthy food purchasing behaviors[35–40], which was consistent with our results using both indices. However, the low correlation between the RFAI and the mRFEI was also expected given the conceptual difference between the two indices and the fact that about 80% of the food retailer visits were beyond residents' home census tracts.

Our results also reveal important racial/ethnic disparities in the food retailer visit patterns. We found that census tracts with predominantly Hispanic populations had both higher mRFEI and RFAI than other groups. This finding is consistent with previous studies showing that Hispanics have better diet quality[41] and access to food[42] than other groups. The higher-than-average mRFEI is possibly due to the rise of ethnic food markets and farmers markets catering to a rising Hispanic population in the US[42]. Further, the higher-than-average RFAI of this group could be partially explained by the immigrant and Hispanic health paradox[43–46], which suggests that first-generation (Hispanic) immigrants in the US exhibit better health behaviors than their native-born peers or the White population, regardless of socioeconomic status (e.g., one recent study found no relationship between fast food access and soda or fruit/vegetable consumption in predominantly Hispanic communities[47]). In addition, conditional on other covariates, residents in predominantly non-Hispanic White census tracts had higher mRFEI and traveled shorter distances for food than others, while predominantly non-Hispanic Black census tracts had lower mRFEI and traveled longer distances. This result is consistent with other previous studies[42,48] and can be partially explained by the lack of food access, especially healthy food retailers in many Black communities. Food retailer visits, however, exhibited an opposite pattern—predominantly non-Hispanic Black census tracts had higher RFAI than predominantly non-Hispanic White census tracts. While this finding still warrants further investigation, it is partially corroborated by several studies finding that non-Hispanic Black populations had non-inferior or better diet quality than non-Hispanic White populations, especially for those with preexisting conditions and after controlling for other sociodemographic factors[49–51]. More importantly, this result signifies the behavioral uncertainties when studying the food environment, and it indicates that food retailer visit behaviors are much more intricate than location-based food environment.

Taken together, these findings posit a pressing need for policy actions that transcend the conventional scope of food environment research, considering human mobility rather than confining the analyses to static residential food context. Over the last two decades, various public policies have been employed to shape the retail food environment, aiming to change food consumers' dietary choices and health outcomes[52]. Geographic access has been a key lever in these efforts, involving measures such as adding a new retailer or extending retailers' operating hours. This geographic policy lever, however, rarely took effect. A systematic review found that 79% of geographic access interventions had a mixed or null effect on diet or health improvement[52]. Our findings further illuminate this challenge by demonstrating that a significant proportion of food retailer visits extend beyond the confines of residents' home census tracts. This insight indicates that altering the immediate community nutrition environment within individuals' local communities may yield only a marginal effect on their food procurement behaviors. In essence,

policies targeted at enhancing food retailers within low-income or segregated neighborhoods may not have a direct influence on the dietary practices of their inhabitants.

Given RFAI's regional delineation, it holds promise as a tool to pinpoint areas that would most benefit from investments in the food environment. Utilizing the RFAI to guide investment allocation could offer more contextualized insights and improve upon the existing USDA designation of "low-income, low-access areas"[29], which have been the priority areas eligible to receive loans, grants, and technical assistance to improve healthy access[53]. These investment initiatives include not only the establishment of traditional grocery stores but also innovative food retail model, including Community Supported Agriculture (CSA), mobile markets, and food co-ops. The RFAI has the potential to redirect focus on areas that may not be traditionally "low-access" yet exhibit unhealthy food retailer visit patterns. Such insights prove pivotal in devising alternative intervention strategies, ranging from health education initiatives to enhancing the affordability of healthy foods and improving transportation infrastructure. As another example, a tool like the RFAI could be a useful component of hospital Community Health Needs Assessments (CHNAs)[54,55]. These assessments, which are required every three years by non-profit hospitals, rely on data that characterizes local community health needs. Measures like the RFAI could serve as invaluable tools for healthcare agencies as they strategize to address the food-related needs of the community.

Nevertheless, this study has limitations. First, translating these findings from an aggregate level into the individual level would be ecologically fallible, as the mobility data were aggregated by the consumers' home census tract and cannot explain distinct food activity patterns at the individual or household scale. This limitation cannot be easily overcome, as the raw mobility data were intentionally collected anonymously without the inclusion of individual characteristics, which serves to protect individual privacy. Second, while SafeGraph sampling is representative of various sociodemographic characteristics and closely corresponds to the US census population counts, especially at the county and state levels, recent studies show that some groups (e.g., Hispanic populations, low-income households, and individuals with low levels of education) might be underrepresented in this data. The extent of this underrepresentation varies across different spatial, temporal, urbanization, and geographic levels[56]. More research is warranted to fully understand the inherent biases and limitations of these large-scale mobility data, as well as to develop methods for overcoming them. Third, due to the observational nature of the data, we would not establish causality. The causal pathway between food activities and cardiometabolic health is a long chain of transformations from health behaviors to health outcomes. Accordingly, it cannot be easily elucidated with aggregate-level data or simple statistical models. To solve this issue, future studies should focus on a smaller geographical region with a longitudinal study design and complement behavioral records with survey data or natural experiments to establish causality. Moreover, the causal pathway to obesity and other CMDs includes tiers of behavioral uncertainties, such as the quality of purchased food and diet practice, which cannot be substantiated by a single aggregate geographic measure such as the one created here. Therefore, when a minority community is targeted for nutrition assistance, it is more imperative to identify structural barriers to healthy diets that are unique to a minority group and to utilize culturally appropriate policy levers acceptable to the community. Fourth, while we followed the definitions of healthy and unhealthy food retailers in the mRFEI as closely as possible when constructing the RFAI, changes in business classification and small differences in the definition (e.g., the employee size of a grocery store was considered in the mRFEI but not in the RFAI) might have contributed to some differences we observed in the two indices. Future research should utilize information from multiple sources (e.g., business information, store

audit data) to more accurately define healthy and unhealthy food retailers. Finally, we intentionally included mobility data from only 2018–2019 to eliminate the impact of the COVID-19 pandemic. Future studies could use more recent data to investigate whether the pandemic has reshaped food procurement activities. The changes experienced during and after the pandemic could pose new challenges to cardiometabolic health.

In conclusion, we utilize large-scale human mobility data on food retailer visits to construct a retail food activity index on the census tract level in the United States. Compared to the traditional food environment measure, the new index exhibits distinctive patterns and has significantly stronger associations with the prevalence of multiple CMDs. The formation of the new index provides an overarching approach to identifying communities with less healthy food procurement activities, signifying their relevance to an uneven landscape of cardiometabolic health. The new index has the potential to be further developed and publicized to inform policy, such as through an interactive analysis platform that is scalable in the US or even applicable to other countries. This effort could eventually guide policy formation and health intervention through both large federal initiatives and tailored community development programs to curb CMDs.

## Methods

This research complies with all relevant ethical regulations and does not involve access to any identifiable private information of human subjects. We conducted a nationwide observational study and developed a new food retailer visit measure, the RFAI, for food retailer visit assessments throughout the US. The development of the tool was reliant on multi-sourced data and the measures listed below.

### Data and measures

For location based food environment index, we obtained the mRFEI from the Centers for Disease Control and Prevention's (CDC) Division of Nutrition, Physical Activity, and Obesity. Since the mRFEI's initial release in 2011, it has been considered the most detailed and comprehensive food environment index in the US[57]. It was measured at the census tract level as a percentage of healthy food retailers in all qualified food retailers using the following formula.

$$mRFEI = \frac{\# \, Healthy \, Food \, Retailers}{\# \, Healthy \, Food \, Retailers + \# \, Less \, Healthy \, Food \, Retailers}$$

(1)

The mRFEI defines the types of food retailers based on their 2007 North American Industry Classification System (NAICS) codes. Healthy food retailers include supermarkets and larger grocery stores (NAICS 445110; supermarkets are stores with ≥ 50 annual payroll employees and larger grocery stores are stores with 10–49 employees), warehouse clubs (NAICS 452910), and fruit and vegetable markets (NAICS 445230; establishments that sell fresh produce and include markets and permanent stands) within census tracts or located ½ mile from the tract boundary. Less healthy food retailers include limited-service restaurants (NAICS code 722211), small grocery stores (NAICS code 445110; the number of employees was three or fewer), and convenience stores (NAICS code 445120) within census tracts or located ½ mile from the tract boundary[13].

To construct retail food activity index, we obtained the residents' food retailer visits in 2018–2019 across the US from SafeGraph's Core Places and Patterns datasets[58]. The data primarily includes anonymized origin-destination (OD) flow data, which were aggregated from about 10% of all GPS-enabled mobile devices in the US. SafeGraph determined a device's "home" by analyzing 6 weeks of data during nighttime hours (between 6:00 PM and 7:00 AM). The home location is defined at the Geohash-7 level (153 × 153-m grid) and is mapped to a census block group, census tract, and country[59,60]. Previous studies showed that

SafeGraph sampling is representative of various sociodemographics (e.g., racial/ethnic composition, education group, and income[60]) and closely corresponds to the US census population counts, especially at the county and state level[61].

For healthy food retailers, we included supermarkets and grocery stores (2017 NAICS code 445110), warehouse clubs (NAICS code 452311), and fruit and vegetable retailers (NAICS code 445230); for less healthy food retailers, we included convenience stores (NAICS code 445120) and limited-service restaurants (NAICS code 722513). Here all supermarkets and grocery stores (regardless of employment size) were categorized as healthy food retailers. This definition is consistent with past literature[17,62,63]. Our final sample included 359,365 food retailer POIs and 94,256,870 visit records in 2018 and 2019, where each record indicated the destination POI, origin (the visitor's home census tract) and the number of visits in each year. The number of included POIs under each NAICS category aligned well with the total number of businesses listed in the 2021 NAICS association statistics (Supplementary Table 1), indicating the comprehensiveness of the POIs under investigation. We further aggregated the data at the census tract level for each origin and calculated various mobility measures, including median distance traveled (the geodetic distances between the POI and centroid of the census tract) and the percentage of total food retailer visits destined within a ½ mile boundary of the home census tracts.

For each home census tract, we further constructed the new index as follows.

$$RFAI = \frac{\# \, visits \, to \, healthy \, food \, retailers}{\# \, visits \, to \, healthy \, food \, retailers + \# \, visits \, to \, less \, healthy \, food \, retailers} \tag{2}$$

The prevalence of CMDs was obtained from PLACES data provided by CDC's Division of Population Health, Epidemiology and Surveillance Branch. The latest PLACES data provides model-based estimates[64,65] at the census tract level across the US based on the Behavioral Risk Factor Surveillance System (BRFSS) data in 2019. In this study, we focused on the prevalence of five CMDs among adults aged over 18: high blood pressure, coronary heart disease, diagnosed diabetes, high cholesterol, and obesity.

For each census tract, we obtained demographic and socioeconomic variables from multiple sources. We obtained the percentile ranking of the SVI in 2018 from CDC (ranging from 0–1, whereas 1 means the most socially vulnerable). The SVI ranks each census tract based on 15 sociodemographic factors derived from the 5-year (2013–2018) American Community Survey (ACS), which are further categorized into four themes: socioeconomic status, household composition and disability, minority status, and housing type and transportation[66]. In addition to the SVI data, we collected representative demographic and socioeconomic variables (many of them were included in the SVI calculation) from the same 5-year ACS data for each census tract[67], including the percentages of the population that are female, minority, low income, have less than a high school education, are under age 5 or over age 64, as well as median family income. To obtain the most accurate demographic characteristics of each census tract including total population, population density, and racial composition, we collected information from the 2020 decennial census[68]. Following previous studies[47], we defined census tracts with predominantly non-Hispanic White, non-Hispanic Black, and Hispanic populations as census tracts with 50% or more of each respective population. We obtained the urban and food desert indicator for each census tract from the 2019 USDA Food Access Research Atlas[29], where a food desert was defined as a low-income census tract where a significant portion of the population lives more than 0.5 miles from a supermarket or large grocery store in urban areas (or 10 miles in rural areas).

## Statistical analysis

We conducted descriptive analyses to summarize the aforementioned measures in Supplementary Table 2. We also performed spatial descriptive analysis and visualizations at the census tract level and summarized the distribution of the distance traveled for food-retailer visits and the percentage of total food-retailer visits within a ½ mile boundary of residents' home census tracts. All spatial visualizations were performed in ESRI ArcGIS Pro 3.0.2[69] and other visualizations were performed in R 4.2.2 using R package "ggplot2" 3.4.3[70,71]. Statistical analyses were performed in STATA 17.0[72]. To explore sociodemographic characteristics that were associated with the distance residents traveled to food retailers (distance from the centroid of the home census tract to the food retailer), we used the percentile ranking of four SVI themes (i.e., socioeconomic status, household composition and disability, minority status, and housing type and transportation), food desert indicator, urban indicator, and population density as the independent variables, and performed multivariate linear regression with median distance traveled (log-transformed) as the outcome and quantile regression with median distance traveled as the outcome at the census tract level. To account for possible coverage bias in the mobility data (i.e., certain areas have more coverage than others), we included per-capita food retailer visits in 2018–2019 (log-transformed) as an additional covariate in each model. To further explore possible racial disparities, we replaced the percentile ranking of SVI theme 3 (minority status) with indicators for census tracts with predominantly non-Hispanic White, non-Hispanic Black, and Hispanic populations as independent variables in alternative model specifications.

To explore sociodemographic characteristics that were associated with the RFAI and mRFEI, we performed multivariate linear regression with the RFAI and the mRFEI as the outcomes in separate models with the aforementioned independent variables at the census tract level.

To explore the independent association between the RFAI or mRFEI and the prevalence of CMDs, we performed multivariate linear regression with each disease prevalence as the outcome and the RFAI or the mRFEI as the key independent variable in separate models. All aforementioned covariates were included, as well as additional sociodemographic variables, including percent of the population who are female, minority, low income, have less than a high school education, are under age 5 or over age 64, median family income, and total number of food retailers (log-transformed) in each census tract.

To ensure the robustness of our results, we also performed several sensitivity analyses: (a) We explored the possible non-linear relationships between the RFAI and each cardiometabolic disease prevalence with the aforementioned covariates using generalized additive models[73]. (b) As the mRFEI was slightly outdated, we checked the robustness of our results by creating another location-based food environment index based on the mRFEI formula using food retailers included in the 2018–2019 SafeGraph data (same as RFAI all supermarkets and grocery stores were categorized as healthy food retailers when constructing this location-based food environment index), and repeated the previous analyses. (c) To further account for the possible scale effect arising from model-based estimates of disease prevalence at the census tract level, we aggregated the data at the county level and repeated the previous analyses. (d) To account for spatial autocorrelations between geographical units, we repeated our analysis at the county level using the spatial error regression framework[74]. (e) To explore whether the relationships vary by racial/ethnic groups, we repeated the previous analysis for each racial/ethnic group separately at the census tract level. We report details in the Supplementary information. Two-sided t-tests were used in all regression analyses and 95% CIs were reported.

## Reporting summary

Further information on research design is available in the Nature Portfolio Reporting Summary linked to this article.

## Data availability

Data used in this study is available at https://dataverse.harvard.edu/dataset.xhtml?persistentId=doi:10.7910/DVN/DMJDVL

mRFEI was obtained from CDC is publicly available and can be accessed here https://stacks.cdc.gov/view/cdc/61367

The prevalence of cardiometabolic diseases data was obtained from PLACES data provided by CDC in 2021. It is publicly available and can be accessed here https://data.cdc.gov/500-Cities-Places/PLACES-Census-Tract-Data-GIS-Friendly-Format-2021-/mb5y-ytti

2018 Social vulnerability index data was obtained from CDC, it is publicly available and can be accessed here https://www.atsdr.cdc.gov/placeandhealth/svi/data_documentation_download.html

2013–2018 American community survey data is publicly available and can be accessed here https://www.census.gov/programs-surveys/acs/data.html

2020 decennial census is publicly available and can be accessed here https://www.census.gov/data/developers/data-sets/decennial-census.html

## Code availability

Code used in this study is available at https://github.com/ranxu-uconn/RFAI.

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

## Acknowledgements
R.X., W.L. and X.C. were supported by a Hatch grant from the College of Agriculture, Health, and Natural Resources, University of Connecticut, funded by the National Institute of Food and Agriculture, United States Department of Agriculture, grant number CONS01031; R.X., D.G., and X.C were supported by the Alan R. Bennett public health policy research funding from College of Liberal Arts and Sciences (CLAS), University of Connecticut and an internal funding for pilot studies addressing U.S. health disparities from the Institute for Collaboration on Health, Intervention, and Policy (InCHIP), University of Connecticut. We appreciate Caitlin Caspi for providing valuable feedback to the study.

## Author contributions
R.X. and X.C. designed the study. R.X. and X.H. designed the methods and performed the analysis. R.X., X.H. and X.C. created the visualizations. R.X., K.Z. and Z.L. organized the data. R.X. and X.C. wrote the manuscript. R.X., X.H., K.Z., W.L., D.G., Z.L., and X.C. revised the manuscript.

## Competing interests
The authors declare no competing interests.
