## [Peer Review File · Nature Communications]

Integrating human activity into food environments can better predict cardiometabolic diseases in the United StatesREVIEWER COMMENTS

Reviewer #1 (Remarks to the Author):

This is a really interesting examination of commercially available data to characterize food environments in a way that incorporates human mobility and food procurement behaviors.

Some sections of the manuscript would benefit from substantial re-writing and editing. The justification for the study in the Intro/Background section is a bit stale; many studies over the past decade have argued that measures need to incorporate human mobility. Few such studies, however, have devised a method to visualize such a measure using discrete administrative boundaries. This in itself seems rather innovative, regardless of the ensuing analysis.

The manuscript requires a great deal of editing for language, including diction (there are some odd word choice), syntax, and voice (it switches between active and passive many times). On occasion, and in a few sections, the authors' meaning is obscured by errors or ambiguities in these aspects.

The results of the analysis are interesting, but somewhat difficult to interpret and understand. The authors might consider splitting this up into a few papers, especially given the number of supplemental figures. The GWR analysis, for example, could maybe stand alone.

Some of the figures could use additional work so the software used to produce them isn't recognizable by default color/font/etc. settings. (See the Stata-produced bar graphs in Figs. S3A-B.)

The Discussion could be more developed for the breadth of analysis presented.

Reviewer #2 (Remarks to the Author):

This paper provides a broad-scale national analysis of the food environment using a newly

created index, the AFEI. There have been several papers over the last decade or so that underscore the importance of considering daily mobility when measuring food exposure. Most measure *exposure* to food options using network buffers or daily activity spaces. This analysis relies on Safegraph data to identify home points for food related points of interest (POI), calculating a score based on actual food procurement behavior. It provides a novel analysis that does contribute to this body of literature, and I believe it should be considered for publication.

That said, there are several points where I would suggest revision. First, the conceptual framework for this index is somewhat underdeveloped. The authors compare AFEI to the mRFEI, but these two measure fundamentally different things. mRFEI measures the food environment itself--what food options are available in a given residential area. AFEI measures food procurement behavior--where residents of a given area purchase food regardless of location. That these two are not correlated is thus not surprising, as they measure different things. This is a crucial distinction in my read, as the AFEI is thus not a food environment measure as suggested in the title, and comparison with the mRFEI just shows that actual behaviors of area residents may be more predictive of health outcomes than the overall food landscape. More importantly, the discussion section speaks very generally of policy interventions, but it's not clear exactly what policies should change based on the AFEI, since we know very little of the reasons *why* behaviors differ across areas--why residents of some areas are more apt to visit healthy retailers than others. More thought needs to be given to the nature of this measure and the policy interventions that it would suggest, especially from a public health perspective.

Similarly, when on p. 6 of the PDF multiple associations are listed between tract demographics and both these indices, there's little explanation of the factors potentially driving this association. The discussion section does address the results for Hispanic areas, but others are not as clearly mentioned. Part of arguing for the value and validity of this measure should include an explanation of why these results fit within existing or emerging frameworks of how food environments and human behavior interact, especially along sociodemographic categories. That is to say, ideally the paper wouldn't just say that non-Hispanic Black populations have a higher AFEI, but also some idea of why that might be the

case--especially when it diverges from the expected value based on the mRFEI.

Another example is the finding on p. 5 that non-Hispanic White tracts had longer travel times than others. From my experience working with these data, this might very well have to do with the combination of suburban sprawl and largely homogenous rural communities in much of the country, but this relationship isn't really investigated. One simple solution would be to include population density and/or a more nuanced urban/suburban/rural classifier. Otherwise, I'd wonder if race is acting as a proxy variable for other factors.

Should these issues be addressed by the authors, I could see value in this article advancing to publication.

Some other smaller notes for revision:

* There are some awkward phrasing/grammar issues throughout the article. For example, in the abstract--"...significant associations iwth multiple cardiometabolic disease prevalence." could be "significant associations with the prevalance of multiple cardiometabolic diseases." The first full sentence on p. 3 of the PDF is quite confusing. "invariably" is used in several points to refer to mRFEIs use, but other indexes, including USDA's, is used. A good proofread would be helpful.

* Along similar lines, the AFEI is referred to as a policy tool, which is confusing to me. It's a tool that conceivably could inform policy, but nothing about it is inherently policy related. Perhaps its an analytical visualization tool?

* Food "foraging" (p. 15) is an odd use of the term, which usually refers to collecting food in the wild or from dumpster diving. Would "procurement" be better?

* On the PDF p. 5, exact distances are used for distance to food retailers, but this is from tract centroids? It's unclear how the precision works here, since tracts can be a mile or more wide, so a median distance with a decimal point seems problematic since there's ambiguity about the distance travelled by each individual.

*Some figures struck me as extraneous. Figure S3 (histograms) doesn't really tell us much and Figure S1 appears to be one of those maps that just shows US population density. Figure S4 has a poor projection of the US and the map itself isn't color blind or black and white printer friendly.

* On p. 8, the effects of one unit increase in AFEI are pretty small (<1%). Could you compare that to the range of observed AFEI values? What does that mean for the 25th vs. 75th percentile tract for AFEI?

* p. 12, this paragraph is quite long and could be broken up.

* It's not clear to me how USDA LILA areas compare to the mRFEI in this article. Don't they measure roughly the same thing? Why are they used in combination, as LILA tracts appear to be an independent variable in your models?

In response to the reviewers' comments, we have incorporated the suggested changes. We hope that these revisions can address the reviewers' concerns and improve the overall quality and rigor of our study.

Reviewer #1 (Remarks to the Author):

R1-1. This is a really interesting examination of commercially available data to characterize food environments in a way that incorporates human mobility and food procurement behaviors.

A1-1: Thank you for your positive comment!

R1-2. Some sections of the manuscript would benefit from substantial re-writing and editing. The justification for the study in the Intro/Background section is a bit stale; many studies over the past decade have argued that measures need to incorporate human mobility. Few such studies, however, have devised a method to visualize such a measure using discrete administrative boundaries. This in itself seems rather innovative, regardless of the ensuing analysis.

A1-2: Thank you. As suggested, we have substantially revised the introduction, expanded the rationale for the study, and highlighted its novelty.

R1-3. The manuscript requires a great deal of editing for language, including diction (there are some odd word choice), syntax, and voice (it switches between active and passive many times). On occasion, and in a few sections, the authors' meaning is obscured by errors or ambiguities in these aspects.

A1-3: The point is well taken. We have carefully revised and proofread the manuscript to mitigate errors and ambiguities. We have also hired a professional editor to proofread the paper in order to meet the highest publishing standard. We hope these efforts have enhanced the manuscript's readability.

R1-4. The results of the analysis are interesting, but somewhat difficult to interpret and understand. The authors might consider splitting this up into a few papers, especially given the number of supplemental figures. The GWR analysis, for example, could maybe stand alone.

A1-4: We have substantially expanded the discussion section and elaborated on the interpretation and implications of the main results. We have also removed the GWR analysis from this manuscript, as suggested.

R1-5. Some of the figures could use additional work so the software used to produce them isn't recognizable by default color/font/etc. settings. (See the Stata-produced bar graphs in Figs. S3A-B.)

A1-5: Thank you. We have improved the qualities of several figures for better clarity and presentation (including Figs. S3A-B (now Fig. S2A-B)).

R1-6. The Discussion could be more developed for the breadth of analysis presented.

A1-6: Thank you. We have substantially expanded the discussion section and elaborated on the interpretation and implications of the main results. Specifically, we have expanded on the discussion of the health disparity and related it to policy implications.

Reviewer #2 (Remarks to the Author):

R2-1. This paper provides a broad-scale national analysis of the food environment using a newly created index, the AFEI. There have been several papers over the last decade or so that underscore the importance of considering daily mobility when measuring food exposure. Most measure *exposure* to food options using network buffers or daily activity spaces. This analysis relies on Safegraph data to identify home points for food related points of interest (POI), calculating a score based on actual food procurement behavior. It provides a novel analysis that does contribute to this body of literature, and I believe it should be considered for publication.

A2-1: Thank you for your positive comment!

R2-2. That said, there are several points where I would suggest revision. First, the conceptual framework for this index is somewhat underdeveloped. The authors compare AFEI to the mRFEI, but these two measure fundamentally different things. mRFEI measures the food environment itself--what food options are available in a given residential area. AFEI measures food procurement behavior--where residents of a given area purchase food regardless of location. That these two are not correlated is thus not surprising, as they measure different things. This is a crucial distinction in my read, as the AFEI is thus not a food environment measure as suggested in the title, and comparison with the mRFEI just shows that actual behaviors of area residents may be more predictive of health outcomes than the overall food landscape.

A2-2: Thank you for your good point. We have now changed our title and renamed this index as retail food activity index (RFAI), as it is conceptually different from the mRFEI. We also highlighted their conceptual and empirical differences throughout the manuscript.

R2-3. More importantly, the discussion section speaks very generally of policy interventions, but it's not clear exactly what policies should change based on the AFEI, since we know very little of the reasons *why* behaviors differ across areas--why residents of some areas are more apt to visit healthy retailers than others. More thought needs to be given to the nature of this measure and the policy interventions that it would suggest, especially from a public health perspective.

A2-3: Thank you for your insightful feedback. The primary objective of our study is to highlight the differences between the new and old location-based indices, showcasing the enhanced value of the former. While delving into the causal pathways of varying behaviors across areas is complex and somewhat beyond the scope of this manuscript, we acknowledged it as a limitation in the discussion section.

In response to your valuable suggestion about policy impacts, we have significantly expanded the discussion section and offered a comprehensive interpretation of our key results in a policy-oriented context, including but not limited to USDA designation of "low-income, low-access areas (L263)" and Community Health Needs Assessments (L271). We believe these revisions have not only improved the overall readability but have also provided a clearer understanding of the study's contribution.

R2-4. Similarly, when on p. 6 of the PDF multiple associations are listed between tract demographics and both these indices, there's little explanation of the factors potentially driving this association. The discussion section does address the results for Hispanic areas, but others are not as clearly mentioned. Part of arguing for the value and validity of this measure should include an explanation of why these results fit within existing or emerging frameworks of how food environments and human behavior interact, especially along sociodemographic categories. That is to say, ideally the paper wouldn't just say that non-Hispanic Black populations have a higher AFEI, but also some idea of why that might be the case--especially when it diverges from the expected value based on the mRFEI.

A2-4: Thank you. Please see our response in A2-3. We have now substantially extended on how and why our new activity-integrated index improves upon the location-based ones for interpreting health disparities when race and ethnicity are involved. Specifically, we have added a new figure (Figure 3) in the result section and we have now elaborated on how our results on racial/ethnic differences of the two indices are consistent with past studies, and possible explanations behind it (L225-246).

R2-5. Another example is the finding on p. 5 that non-Hispanic White tracts had longer travel times than others. From my experience working with these data, this might very well have to do with the combination of suburban sprawl and largely homogenous rural communities in much of the country, but this relationship isn't really investigated. One simple solution would be to include population density and/or a more nuanced

urban/suburban/rural classifier. Otherwise, I'd wonder if race is acting as a proxy variable for other factors.

A2-5: Very good point and we agreed. To that end, we have now included population density in our analysis, as suggested. While the inclusion of population density did not change most of our results (e.g., factors explaining RFAI/mRFEI or cardiometabolic health outcomes), it was one of the strongest predictors for distance traveled to food retailers, and it did change our results/interpretation on the racial group differences in distance traveled to food retailers, i.e., after controlling for population density and other covariates, residents in predominantly white census tracts traveled shorter in distance for food than others while those in predominantly black census tracts traveled longer in distance. We have now noted and further discussed this in the discussion section (e.g., L215 and L234).

Should these issues be addressed by the authors, I could see value in this article advancing to publication.

Some other smaller notes for revision:

R2-6. * There are some awkward phrasing/grammar issues throughout the article. For example, in the abstract--"...significant associations iwth multiple cardiometabolic disease prevalence." could be "significant associations with the prevalance of multiple cardiometabolic diseases." The first full sentence on p. 3 of the PDF is quite confusing. "invariably" is used in several points to refer to mRFEIs use, but other indexes, including USDA's, is used. A good proofread would be helpful.

A2-6: Thank you. In this revision, we have carefully revised and proofread the manuscript to mitigate errors and ambiguities. We have also hired a professional editor to proofread the paper in order to meet the highest publishing standard. We hope these efforts have enhanced the manuscript's readability.

R2-7. * Along similar lines, the AFEI is referred to as a policy tool, which is confusing to me. It's a tool that conceivably could inform policy, but nothing about it is inherently policy related. Perhaps its an analytical visualization tool?

A2-7: Thank you. We have now removed this language and say it's a useful tool that can inform policy.

R2-8. * Food "foraging" (p. 15) is an odd use of the term, which usually refers to collecting food in the wild or from dumpster diving. Would "procurement" be better?

A2-8: Done.

R2-9. * On the PDF p. 5, exact distances are used for distance to food retailers, but this is from tract centroids? It's unclear how the precision works here, since tracts can be a mile or more wide, so a median distance with a decimal point seems problematic since there's ambiguity about the distance travelled by each individual.

A2-9: Correct. If we use integer values to represent distance, median or mean can still have decimal places. So we have decided to keep the decimal places to be consistent with other numbers reported in the manuscript. We have now clarified in multiple places in the manuscript that distance is defined as the distance from the centroid of the census tract to a food retailer. We have also noted this in the limitation section (i.e., mobility data used in this study were aggregated by the consumers' home census tract and cannot explain the distinct food activity patterns at the individual or household scale).

R2-10. *Some figures struck me as extraneous. Figure S3 (histograms) doesn't really tell us much and Figure S1 appears to be one of those maps that just shows US population density. Figure S4 has a poor projection of the US and the map itself isn't color blind or black and white printer friendly.

A2-10: Thank you and points are well taken. Given they are all in the supplementary information (which does not have a length limit and will be separated from the main manuscript when published) that will only appear online, we have decided to keep most of them or consolidate the figures (now 9 figures in the supplementary information instead of 11 in the original submission) to give readers more information. We have improved the quality of several figures. All cartographical issues noted by the reviewer have been corrected.

R2-11. * On p. 8, the effects of one unit increase in AFEI are pretty small (<1%). Could you compare that to the range of observed AFEI values? What does that mean for the 25th vs. 75th percentile tract for AFEI?

A2-11: Good point. In the main text and Figure 4, we have now changed the interpretation to “estimated change in the prevalence of five cardiometabolic diseases associated with one interquartile increase in RFAI or mRFEI (i.e., 25 percentile to 75 percentile).” While the RFAI index was not designed to maximize its association with the prevalence of cardiometabolic diseases, across models we found that the associations between RFAI and obesity were about three times stronger than that between mRFEI and obesity. We point out this finding in the discussion.

R2-12. * p. 12, this paragraph is quite long and could be broken up.

A2-12: Thank you. We have rewritten the discussion and broken up long paragraphs.

R2-13. * It's not clear to me how USDA LILA areas compare to the mRFEI in this article. Don't they measure roughly the same thing? WHY are they used in combination, as LILA tracts appear to be an independent variable in your models?

A2-13: Thank you. These two are conceptually different - LILA areas consider both low income populations and low access to supermarkets or large grocery stores, while mRFEI only focuses on food retailers but includes more store types such as small grocery stores, fruit and vegetable markets, convenience stores and fast food restaurants. Empirically, LILA area is a binary variable and mRFEI is a continuous variable (0-100, representing the percentage of healthy food retailers in a census tract). Due to these reasons, the correlation between the two was also very low (-0.09) on the census tract level. And given USDA LILA areas are often used in designing policies, we have decided to keep this variable in our model, and we have further clarified this in the methods section.

REVIEWERS' COMMENTS

Reviewer #1 (Remarks to the Author):

This manuscript is substantially improved in terms of language/writing, but some issues regarding clarity and focus remain.

Some specific items that should be considered:

Figure 1 presents an almost completely random pattern for a measure that, in the end, is not really important to the main findings. Is that really worth showing above all the other supplemental maps/figures?

While it escaped my notice during initial review, I find it a bit strange that the definitions for "healthy" and "less healthy" retail food locations vary between mRFEI and RFAI indices. For example, in calculating mRFEI the authors include fast food (NAICS 722211) in "less healthy", but switch to limited service resaurants (NAICS 722511) for RFAI. Also, groceries and supermarkets (NAICS 445110) were handled differently. These differences should be explicitly justified in the Methods and the implications of these differences should be covered in the Discussion, since they alone could account for some of the differences in mRFEI and RFAI.

Figure 3 is confusing despite its relative simplicity. It does not "stand on its own". In other words, the reader must carefully digest the text to understand what the figure is showing. Even then, I think it is a bit challenging. Consider that Figure 3 makes no reference to the Social Vulnerability Index, despite that measure being mentioned in the sentence first referencing Figure 3 (lines 153-154). Furthermore, the title/caption of Figure 3 describes a comparison of "racial/ethnic group" and "other census tracts", which is not quite what the text says. Please consider improving the design and clarity of Figure 3 (and its caption/title).

Reviewer #2 (Remarks to the Author):

I appreciate the thoughtful and thorough revisions undertaken by these authors, both in

reconceptualizing the relationship between mRFEI and the RFAI. The expanded discussion section is also much stronger than the previous draft. I have no further revisions to suggest.

We have prepared a detailed response to all comments and made corresponding revisions in the manuscript.

REVIEWERS' COMMENTS

Reviewer #1 (Remarks to the Author):

This manuscript is substantially improved in terms of language/writing, but some issues regarding clarity and focus remain.

Some specific items that should be considered:

R1-1. Figure 1 presents an almost completely random pattern for a measure that, in the end, is not really important to the main findings. Is that really worth showing above all the other supplemental maps/figures?

A1-1. Thank you. As we described in the results section, Figure 1 shows that the majority of the food retailer visits were beyond residents' immediate neighborhoods. On average, only 20.8% (standard deviation [SD] = 13.9%) of food retailer visits were within a half-mile boundary of the residents' home census tracts. This is crucial in motivating our study and justify the needs of integrating human activity into the study of food environments. So we decided to keep Figure 1 and we have now further clarified what Figure 1 conveys in the introduction.

R1-2. While it escaped my notice during initial review, I find it a bit strange that the definitions for "healthy" and "less healthy" retail food locations vary between mRFEI and RFAI indices. For example, in calculating mRFEI the authors include fast food (NAICS 722211) in "less healthy", but switch to limited service resaurants (NAICS 722511) for RFAI. Also, groceries and supermarkets (NAICS 445110) were handled differently. These differences should be explicitly justified in the Methods and the implications of these differences should be covered in the Discussion, since they alone could account for some of the differences in mRFEI and RFAI.

A1-2. Thank you. When defining healthy and unhealthy food retailers in RFAI we followed the operationalization of mRFEI as closely as possible. For example, in both RFAI and mRFEI all limited-service restaurants were defined as unhealthy food retailers - mRFEI used 2007 NAICS code 722211, which is equivalent to 2017 NAICS code 722513 used in the RFAI.

The only difference is the employee size of a grocery store was considered in the mRFEI but not in the RFAI. We choose to simplify the definition in the RFAI as many past studies (e.g., Ball et al. 2009, Chen 2017, and Horowitz et al. 2004) consider healthy stores to be green retailers, or those carrying fresh produce and vegetables, without the component of employee size. Also, studies have shown that small neighborhood grocery stores serve the community needs of healthy food, especially in rural areas, where chained stores are rare (Sharkey 2009).

On a separate note, given that stores with fewer than or equal to 3 employees constituted a small portion of the data, excluding them will not significantly alter our main findings that RFAI is a better predictor of cardiometabolic diseases than mRFEI.

However, we also agree with the reviewer that this small difference in the definition could give rise to some differences in the index patterns. For this reason, we have now further clarified this difference in definition in the methods section, and also included it as a limitation in the discussion section. We also call for better defining healthy and unhealthy food retailers based on information from multiple sources (e.g., business information, store audit data) in the manuscript.

References:

Ball, K., Timperio, A., & Crawford, D. (2009). Neighbourhood socioeconomic inequalities in food access and affordability. *Health & Place*, 15(2), 578-585.

Chen, X. Take the edge off: a hybrid geographic food access measure. *Applied Geography* 87, 149–159 (2017).

Horowitz, C. R., Colson, K. A., Hebert, P. L., & Lancaster, K. (2004). Barriers to buying healthy foods for people with diabetes: evidence of environmental disparities. *American Journal of Public Health*, 94(9), 1549-1554.

Sharkey, J. R. (2009). Measuring potential access to food stores and food-service places in rural areas in the US. *American Journal of Preventive Medicine*, 36(4), S151-S155.

R1-3. Figure 3 is confusing despite its relative simplicity. It does not "stand on its own". In other words, the reader must carefully digest the text to understand what the figure is showing. Even then, I think it is a bit challenging. Consider that Figure 3 makes no reference to the Social Vulnerability Index, despite that measure being mentioned in the sentence first referencing Figure 3 (lines 153-154). Furthermore, the title/caption of Figure 3 describes a comparison of "racial/ethnic group" and "other census tracts", which is not quite what the text says. Please consider improving the design and clarity of Figure 3 (and its caption/title).

A1-3. Thank you and good point. We have now (a) changed the labels in Figure 3, (b) edited both the text and the figure caption to improve the clarity of Figure 3.

Reviewer #2 (Remarks to the Author):

R2-1. I appreciate the thoughtful and thorough revisions undertaken by these authors, both in reconceptualizing the relationship between mRFEI and the RFAI. The expanded discussion section is also much stronger than the previous draft. I have no further revisions to suggest.

A2-1. Thank you!